# Barrier Membrane in Regenerative Therapy: A Narrative Review

**DOI:** 10.3390/membranes12050444

**Published:** 2022-04-20

**Authors:** Muhammad Syafiq Alauddin, Nur Ayman Abdul Hayei, Muhammad Annurdin Sabarudin, Nor Haliza Mat Baharin

**Affiliations:** 1Department of Conservative Dentistry and Prosthodontics, Faculty of Dentistry, Universiti Sains Islam Malaysia, Kuala Lumpur 55100, Malaysia; 2Department of Periodontology and Community Oral Health, Faculty of Dentistry, Universiti Sains Islam Malaysia, Kuala Lumpur 55100, Malaysia; nurayman@usim.edu.my (N.A.A.H.); annurdin@usim.edu.my (M.A.S.); halytzar@usim.edu.my (N.H.M.B.)

**Keywords:** bone regeneration, tissue scaffolds, guided tissue regeneration, periodontal, dental implants

## Abstract

Guided bone and tissue regeneration remains an integral treatment modality to regenerate bone surrounding teeth and dental implants. Barrier membranes have been developed and produced commercially to allow space for bone regeneration and prevent the migration of unwanted cells. Ideal membrane properties, including biocompatibility, sufficient structural integrity and suitable shelf life with easy clinical application, are important to ensure good clinical regenerative outcomes. Membranes have various types, and their clinical application depends on the origin, material, structure and properties. This narrative review aims to describe the currently available barrier membranes in terms of history, main features, types, indication and clinical application and classify them into various groups. Various membranes, including those which are resorbable and non-resorbable, synthetic, added with growth factors and composed of modern materials, such as high-grade polymer (Polyetheretherketone), are explored in this review.

## 1. Introduction

Various barrier membranes have been used generally in dentistry to complement bone augmentation in implant therapy and periodontal regenerative dentistry. Guided bone regeneration (GBR) and guided tissue regeneration (GTR) have been explored extensively and accepted clinically as a core procedure to regenerate the loss of periodontal tissue [1,2]. Conventionally, a barrier membrane is implanted on a regenerative area which has lost its volumetric tissue to prevent the migration of undesired cells from the gingival epithelium and connective tissue [3,4]. Ideally, the implanted barrier membrane provides a shielding effect for up to 6 weeks and approximately 24 weeks for periodontal tissue regeneration and bone augmentation therapy, respectively. Thus, this membrane provides the desired space for tissue regeneration and ultimately pre-selectively guides periodontal ligament cells and bone regeneration [5,6,7].

Ideal membrane properties are important to ensure good clinical regenerative outcomes. Such properties include biocompatibility and non-toxicity to the surrounding tissue and the body of the organism; high tissue tolerance to ensure progressive and complete integration with the periodontal fibers; adequate structural integrity and good dimensional stability (i.e., ability of the membrane to maintain its shape and position until degradation or removal); tolerable clinical handling, along with suitable storage time; simple application and modification with a tack pin or sutured through; selective permeability to prevent the invasion of epithelial cells, while promoting the proliferation of osteogenic cells; space maintenance for regenerative cells; and adequate blood-clot formation to enhance angiogenesis and vascularity for regeneration. The criteria for ideal regenerative procedure are described as “PASS”, which consists of primary, non-tension wound closure that enables healing by primary intention, angiogenesis to promote blood supply to the regenerative area, stability of clot to allow development and proliferation of osteogenic cell and space maintenance for undifferentiated mesenchymal cells platform [8,9,10].

## 2. Materials and Methods

This paper aims to summarize recent biomaterials and contemporary membranes utilized in periodontal regeneration and implant therapy in the market. A literature search was performed in electronic databases, including PubMed, Medline, OVID and Web of Science, by using the following keywords: “bone augmentation”, “guided bone regeneration”, “GBR”, “guided tissue regeneration”, “GTR”, “alveolar ridge preservation” and “barrier membranes”. Documents published in English were selected, and the articles were further screened to identify their relevance to this review. This narrative review briefly discusses the types and classification of biologically and synthetically derived barrier membranes, as well as the clinical indications and considerations to utilize them and their respective advantages and disadvantages.

## 3. Resorbable Membranes

Several resorbable membranes have been developed and proved clinically effective in the management of periodontal and peri-implant defects. The nature of these membranes being resorbable prevents the need for second surgery. Thus, they are preferred by patients over non-resorbable membranes. For the past 10 years, commercially available resorbable membranes have been used to treat periodontal and peri-implant defects through GTR, based on epithelial exclusion principle [11].

### 3.1. Collagen Membranes

Collagen membranes are natural, resorbable membranes made from human, porcine or bovine sources, such as pericardium, dermis and Achilles’ tendon. Type I collagen is abundant in the periodontal connective tissue and, thus, has been widely used to develop commercial collagen membranes. The attractive properties of collagen membranes include biocompatibility, hemostatic and chemotactic support and wound-healing enhancement through clot stabilization. The capability of collagen membranes to prevent epithelial downgrowth and weak immunogenicity has made them suitable for periodontal regeneration.

However, early resorption may reduce periodontal regeneration capacity. Therefore, various crosslinking techniques have been developed to prolong the absorption time and achieve excellent periodontal regeneration. These techniques include the use of glutaraldehyde, formaldehyde, ultraviolet light, hexamethylenediisocyanate and diphenylphosphorylazide. The resorption time for non-crosslinked collagen membranes ranges from 5 days to 28 days [12,13], whereas crosslinked membranes remain intact after 14 days. This resorption time is sufficient to prevent epithelial downgrowth during early wound healing within the first 14 days. In addition, the higher the crosslinking degree of a specific collagen membrane, the slower the resorption degree of the material itself. However, the prolonged existence of the membrane does not favor regeneration during wound healing. A double-blinded randomized controlled trial found that highly crosslinked collagen prolongs the resorption rate and negatively affects regeneration in case of membrane exposure [14]. Tissue dehiscence is significantly higher in highly crosslinked collagen membranes than in native collagen membranes (*p* = 0.0455) [14]. These findings suggest that a membrane barrier is needed only during the early phase of wound healing. Bunyaratavej and Wang had comprehensively elaborated cross-linking techniques to retard the degradation rates of collagen membranes [15]. 

Adequate biological space and time are required to allow bone and periodontal ligament cells to repopulate the wounded area. Excellent clinical results have been achieved with the use of collagen membranes in conjunction with bone graft. Bone graft not only has osteoinductive capability but also allows space maintenance for bone cells to repopulate the area. Collagen has been combined with other materials, such as collagen membrane, fibronectin and heparan sulfate, to enhance space maintenance and recruit cells with regenerative potentials.

Periodontal regeneration during wound healing requires the attraction of periodontal ligament fibroblasts to regenerate new periodontal ligament, new cementum and new bone. Clot stabilization and chemotaxis toward fibroblasts increase periodontal regeneration. Biocompatibility enhances wound healing and favors regeneration. A previous study compared collagen with expanded polytetrafluoroethylene (ePTFE) in terms of biocompatibility and found that the latter inhibits gingival fibroblast synthesis, whereas the former enhances cell proliferation [16]. Another study performed an enzyme-linked immunosorbent assay and found no specific immunoreaction against collagen [17].

### 3.2. Clinical Evidence

The performance of collagen membranes in the management of intrabony periodontal defects through regeneration has been recognized since the late 1980s. A probing pocket depth reduction as high as 4 mm has been reported following GTR procedures using collagen membranes [18]. Cortellini et al. compared the clinical attachment level gain after GTR and access flap surgery in intrabony defects [18]. Greater attachment level gain was observed after GTR, using resorbable membranes, compared with access flap alone. The results may be attributed to several factors, such as the crosslinking technique, width of intrabony defect, initial probing depth and measurement technique.

Furthermore, collagen matrix (Mucograft) infused with recombinant human platelet-derived growth factor BB (rhPDGF-BB) effectively increases gingival thickness prior to anterior implant prosthesis fixation [19]. Initially, the edentulous area significantly lacks bone height. Thus, GBR was performed by using titanium-reinforced ePTFE and bone graft. After 6 months, sufficient bone height and volume were achieved, and the implant was placed. A collagen membrane infused with rhPDGF-BB was placed over the implant, and sufficient healing time was allowed prior to tissue thickness measurement. The gain in the transmucosal distance was significant with tissue-thickness gain measured at apical, 0.87 mm; central, 2.14 mm; and occlusal, 0.35 mm [19].

### 3.3. Fibrin

Wound healing involves clearing bacterial infection through leukocytes and tissue formation through the attraction of fibroblasts. The process is concurrent with angiogenesis, which accelerates healing by increasing the supply of leukocytes and growth factors. The understanding of wound healing has led to the use of platelet concentrates to enhance perfect wound healing and increase the degree of regeneration. Platelet-rich fibrin mimics natural wound healing and amplifies it when the blood supply is deemed insufficient. Autologous platelet concentrates (APCs) are further discussed in a subsection below.

### 3.4. Placenta

Other natural biomaterials that have recently gained attention are chorion membrane (CM) and amniotic membrane (AM) derived from human placenta. When used in oral soft tissue management, these membranes secrete anti-inflammatory cytokines, growth factors and chemokines and exert antimicrobial effects. They also have low immunogenicity and improve epithelization. Gulameabasse et al. recently published a systematic review of 21 studies conducted on 375 human patients on the use of CM and amnion/chorion membrane (ACM). They found that CM and ACM are effective alternatives to current techniques in treating various oral soft-tissue defects, including gingival recession, intrabony and furcation defects, alveolar ridge preservation, keratinized tissue width augmentation around dental implants, maxillary sinus repair and large bone reconstruction [20]. However, further studies are necessary to investigate their role in bone regeneration.

In an allograft material, cross-infection is an integral issue. Several processing methods of CM and ACM prior to use have been documented. These methods include freeze-drying [21,22], decellularization and freeze-drying [23], de-epithelization and/or dehydration [24] and gamma irradiation [25]. Gamal et al. used the ACM for furcation management and found that it improves furcation defect and bone quality and promotes osseointegration [24]. The osteogenic potential is attributed to the ability of ACM to recruit progenitor cells. In addition, CM and ACM exert analgesic property through close adaptation to bone defects and coverage of nerve endings. Several studies reported lower pain scale when using CM compared with other membranes [26,27]. CM and ACM are resorbable, thereby preventing a second surgical procedure. However, few studies investigated the resorption time for placental membranes.

### 3.5. Chitosan

Chitosan is a deacylated chitin derivative which is biocompatible, self-resorbed and has antimicrobial properties. It exerts osteoinducing effect, acts as a hydrating agent and enhances tissue healing. A laboratory test on human periodontal ligament cells showed that composite membranes composed of chitosan and bioactive glass promote cell metabolic activity and mineralization. Chitosan is a potential candidate for GTR [28]. However, the applications of chitosan are limited by its low biodegradation, poor mechanical properties and ineffective hemostasis maintenance. Electrospinning and lyophilization improve the properties of membranes as effective scaffolds [29,30]. Chitosan-infused membranes prepared by using electrospinning produce an aligned and random fiber morphology with a surface conducive to cellular attachment. The fibers support matrix deposition, and the surface layer prevents junctional epithelium, thereby maintaining space for periodontal regeneration [30]. Zhang et al. generated a unique multifunctional scaffold by combining chitosan, polycaprolactone and gelatin through electrospinning and lyophilization and then implanted the membrane subcutaneously; the results reveal that the membrane has low immunogenicity, its degradation rate resembles tissue regeneration and it prevents external cell invasion [29]. Nevertheless, comprehensive clinical studies are needed prior to the recommendation of this membrane for clinical use. Chitosan is a potential candidate for affordable and low-cost GTR biomaterials in the future.

Nonetheless, collagen membranes can be degraded by collagenase if exposed to bacterial colonization. Several periodontal pathogens, such as *Porpyromonas gingivalis* and *Bacteroides melaninogenicus*, produce collagenase, an enzyme that degrades membranes prematurely [31]. Metronidazole-impregnated collagen has been developed to enhance wound healing through its antibacterial effect. In addition, the membrane must be secured from exposure to the oral environment to ensure minimal bacterial colonization. Primary closure of the surgical site is crucial for effective regeneration using collagen membranes.

Achieving primary closure may be difficult in cases of insufficient keratinized tissue and bone defects, because of periodontitis and peri-implantitis. Coronally advanced flap, vertical mattress suture and tissue punch technique are widely used to achieve primary closure. Regeneration will be negatively affected upon membrane exposure in GTR procedure [32].

Another disadvantage of resorbable membranes is their inability to maintain adequate space unless the defect morphology is favorable. Collagen membranes are vastly used to treat intrabony periodontal defects through GTR. Three-wall defect morphology is required for the successful GTR of defect widths not more than 37° to the tooth axis [18]. GBR procedures were performed to treat horizontal and vertical bone defects. The use of tenting screws is recommended to reduce membrane mobility and achieve adequate space. Titanium-reinforced, pin-reinforced and non-resorbable membranes are alternative materials in GBR.

### 3.6. Current Development of Resorbable Membranes

The porcine-derived collagen bioactive membrane CelGro^TM^ (Orthocell Ltd., Murdoch, Australia) was developed for GBR in dental and orthopedic applications [26]. CelGro^TM^ promotes vascularization [33], induces cellular recruitment [34] and upregulates pro-osteogenic factors at the implant site [35]. Compared to with the commercially available collagen membrane Bio-Gide^®^, CelGro^TM^ shows much better cortical alignment and lower porosity at the defect interface. Celgro^TM^ can restore bone defects without complications or adverse events. Cone-beam computed tomography (CBCT) images show significantly increased bone formation horizontally and vertically, which provides sufficient support to the implants within 4 months [36].

Collagen membranes can modulate the osteoimmune response of macrophages. Chen et al. modified a collagen membrane by coating it with a nanometer bioactive glass (hardysonite) through pulsed laser deposition for GBR and evaluated its ability to enhance osteogenesis through osteoimmunomodulation [37]. They found that the modified collagen membrane can enhance the osteogenic differentiation of bone-marrow-derived mesenchymal stem cells, suggesting that collagen membranes with nanometer-sized hardysonite coating are promising for GBR applications. In addition, Annen et al. developed a collagen membrane with prolonged resorption time to overcome early resorption limitation. However, the results showed significantly higher membrane exposure in the new collagen membrane than in the native collagen membrane [14].

## 4. Non-Resorbable Membranes

Cellulose acetate (CA) was the earliest material used in non-resorbable membranes, which are intended to keep the gingival connective tissue away from the root surface and allow periodontal regeneration [38]. CA has been used because of its outstanding properties, including neutrality, biocompatibility, low cost and renewability [39]. Non-resorbable membranes, which can be further classified into metal, ePTFE and dense PTFE (dPTFE) with or without titanium reinforcement, are widely used in periodontal regenerative approaches, such as GTR/GBR, are collectively depicted in Figure 1. GTR/GBR requires a membrane that works as a physical barrier that can prevent the competitive invasion of highly proliferative cells of the surrounding tissue, mainly fibroblasts and epithelial cells. Meanwhile, the native cell proliferation properties of the natural regeneration region should be promoted [40,41].

Non-resorbable metal-based membranes can be subclassified into titanium mesh and titanium foil. Titanium mesh was first created in the 1960s as a vital-organ restraint device in trauma patients and for reconstructive applications in oncology patients as a device to constrain and immobilize particle autogenous bone grafts received from extraoral locations. In the 1980s, their use was expanded to include bone augmentation to allow for the implantation of dental implants [42]. Titanium mesh has good mechanical qualities, including high strength and stiffness, which provide space for osteogenesis; in addition, its stability preserves bone graft volume during wound healing, and its elasticity can minimize oral mucosa constriction [43]. Given its good flexibility, titanium mesh can adapt to various bone abnormalities. Horizontal and vertical bone augmentation can be achieved in processes with delayed or simultaneous implantation, using titanium mesh, which has strong osteogenesis prediction. In a recent study, eight implants were treated by using preformed titanium mesh; after a 12-month follow-up, they reported a mean horizontal bone gain of 4.95 ± 0.96 mm and a mean horizontal thickness of the buccal plate of 3.25 ± 0.46 mm clinically [44]. Nonetheless, the titanium mesh may be effective in supporting bone regeneration at the dehiscence area, but the exposure of the titanium mesh remains an issue. In addition, its rigidity and the sharp edges created by trimming and contouring might irritate the mucosa and are associated with an increased risk of membrane exposure [45]. Titanium foil prepared as preformed mesh could be advantageously used in large areas of ridge defects because of its excellent properties, such as stiffness, biocompatibility, non-permeability and customizability for GBR [46]. Currently, dPTFE has been proposed for use in regions with large ridge atrophies to prevent graft contamination in the case of undesired exposure of the device [47]. Unfortunately, the stiffness of these non-resorbable membranes, which can be increased with simultaneous implant placement, can be compromised by the chewing function. In addition, the adaptation and fixation of the device to the recipient site is time-consuming [48]. Therefore, the titanium foil membrane can be an alternative to overcome the drawbacks of dPTFE membranes, maximize the treatment outcome and simplify the surgical phase [46]. In a previous study, the success rate of using titanium foil as a membrane barrier was 88.2%, and the average peri-implant bone reabsorption was 1.17 ± 0.41 mm. The average rate of graft contraction was 19.4% ± 10.55%. Meanwhile, the mean percentage occupied by mineralized bone was 48.03% ± 5.93%, whereas those of bone marrow and graft material were 36.1% ± 2.81% and 15.87% ± 4.87%, respectively [46]. However, a long follow-up study regarding the use of titanium foil as a membrane barrier to protect the graft is currently lacking.

Overall, ePTFE has shown positive results in regenerative procedures, provided it has primary closure [49]. An ePTFE membrane features a chemically stable, biocompatible, inert polymer, but its structure is porous and flexible. These properties enable this membrane to resist degradation produced by microbiological or enzymatic reactions [49,50]. These membranes consist of two distinct parts: an open microstructure (100–300 µm porosity) and an occlusive structure (<8 µm porosity). The porous microstructure stimulates the ingrowth of collagen fibrils, thereby improving membrane stability and facilitating nutrient transport through its pores, which, together, stimulate new bone formation during the first healing period [51]. By contrast, the occlusive component is generally impervious to fluids and prevents soft tissue cells from migrating into the area of bone development [52]. As a result, barrier materials should have a porous fraction to generate optimal regenerative therapy results [51,52,53]. However, premature exposure of ePTFE membranes is relatively common and is reportedly approximately 30–40%, and this may lead to infection and lack of new bone formation as a result of fibrous tissue ingrowth [54]. Therefore, primary closure is deemed necessary over ePTFE membranes, but this can be challenging in larger defects [55]. The need for additional surgery to remove the membrane increases the risk of exposing newly regenerated bone to bacteria. The timing of membrane removal is also important, because early removal can lead to resorption of regenerated bone, whereas late removal can increase the risks of bacterial contamination and infection [56]. The conventional product available worldwide with the types of material are briefly describe in Table 1.

The smooth outward face of dPTFE prevents tissue ingrowth, but this can lead to poor socket flap adhesion and tissue dehiscence [57]. A dPTFE membrane has a high density and smaller pore size (0.2 μm), preventing bacterial infiltration and leading to lower risks of infection when exposed [55]. In addition, primary closure over the membrane is not necessary [52]. A previous randomized controlled trial study found no significant difference in bone regeneration between e-PTFE and d-PTFE at 6 months after operation [52]. In the advancement of PTFE-based membranes, a rigid structure with titanium reinforcement was incorporated into these biomaterials [56]. TR membranes, such as the TR ePTFE membrane Gore-Tex^®^, which contains a titanium frame inside, can be shaped into a desirable form [52]. Meanwhile, a TR dPTFE membrane (Cytoplast™), with increased mechanical integrity, can enhance bone-graft stabilization, while also occluding soft tissue [58,59]. A clinical example of the utilization of non-resorbable membrane are depicted in Figure 2.

## 5. Synthetic Membranes

Synthetic biodegradable membranes are made of polymers. Natural polymers, such as collagen, are readily biocompatible, and the protein may enhance cell adhesion and proliferation. However, the mechanical and physical properties of natural polymers are inferior to those of synthetic polymers. For instance, natural polymers have less tensile strength and a higher degradation rate than synthetic polymers [60]. Synthetic polymers are biocompatible, making them suitable as biomaterials. In addition, synthetic polymers have tailorable physiochemical properties according to the desired outcomes and manufacturing reproducibility [61,62,63]. Some synthetic polymers are degradable, and they have gained much attention for membrane development in tissue engineering, because secondary surgery is not needed to remove the membrane.

Biodegradable synthetic polymers to be used as biomaterials for tissue engineering should have several advantageous properties. For instance, they should be biocompatible and not induce inflammatory changes around the tissue. They should also be degraded when the surrounding tissue is ready to function. Excellent physiochemical properties, according to the intended outcome, are also required. Therefore, the current research in biodegradable synthetic polymers is attempting to synthesize polymers and copolymers that can match the ideal properties of the desired function of the biomaterial.

Synthetic biodegradable membranes for biomedical and tissue engineering have several types, including polylactic acid (PLA), polyglycolic acid (PGA), polycaprolactone (PCL), poly(glycolide-co-lactide) copolymer and other copolymers. The investigation of other types of biodegradable synthetic polymers and copolymers is still underway.

Among these polymers, PLA and PCL are the most commonly investigated for tissue engineering. PLA has been studied since the 1980s, and its physiochemical properties have been improved to suit the requirement of an ideal membrane. Since then, the use of PLA has been greatly increasing, especially in GTR and GBR [61]. PLA has four types: poly(L-lactic acid) (PLLA); poly(D-lactic acid) (PDLA); poly(D,L-lactic acid) (PDLLA), a racemic mixture of PLLA and PDLA; and meso-poly(lactic acid). For biomaterial use, PLLA and PDLLA are widely studied because of their excellent properties [62]. GUIDOR^®^ Matrix Barrier is a commercially used PLA membrane made from the combination of PLLA and PDLLA for periodontal regeneration.

Despite being biocompatible as a biomedical device, PLA releases acidic degradation products that can induce inflammation [60,64]. In addition, PLA undergoes slow degradation, which may even take a year [61,62,64]. This disadvantage may induce inflammation at the regeneration site. However, it may also allow another functional property to be added, such as drug and growth factor release [61].

PCL is another type of polymer that is widely investigated for tissue engineering. In contrast to PLA and PGA, PCL does not produce acidic degradation products [60,61,62]. Therefore, it is commonly used for biomedical purposes. However, the degradation rate of PCL is longer than that of PLA and PGA; its complete resorption may take up to 3 years [61]. Among PLA, PGA and PCL, PCL has the lowest tensile strength, tensile modulus and melting temperature [65]. The hydrophobicity of PCL is also a disadvantage. Therefore, PCL is commonly combined with other polymers before it can be used as a biomaterial. Previous studies combined PCL with gelatin, a natural polymer, to improve its mechanical, physical and chemical properties [66,67].

PGA is one of the first biodegradable polymers studied for biomaterial applications. However, it has poor mechanical strength because of its rapid degradation. Similar to PLA, it also produces acidic degradation products that may induce undesired inflammatory response [60,62].

Synthetic polymers can undergo various modifications. Although these polymers are not naturally osteoinductive, additional properties, such as drug delivery and growth factor release, can be added to enhance their properties. Considering the advantages and disadvantages of biodegradable synthetic polymers, additional works are warranted to develop the most ideal type of biodegradable polymer that may serve as an excellent barrier membrane.

## 6. Autologous Platelet Concentrate (APC)

### Types of Autologous Platelet Concentrate

The challenge in GTR involves the replacement and reconstruction of massive tissue defects, especially in the presence of local and systemic contributing factors, such as habitual smoking, diabetes mellitus and multi-walled defects. Utilizing grafting materials, membrane barriers and additional therapy of biologic agents will be beneficial to regenerate the desirable amount of defect quality and quantity. In challenging cases, additional biologic agents will help promote healing induction and conduction in the local surgical area, making the healing process predictable and faster to produce true periodontal regeneration [68,69]. APC has various types, including pure platelet-rich fibrin (PRP), leukocyte platelet-rich fibrin, advanced platelet-rich fibrin (A-PRF), injectable platelet-rich fibrin, titanium platelet-rich fibrin, prepared platelet-rich lysate and concentrated growth factor (CGF). PRP was initially developed for medical purposes, such as the management of severe thrombopenia, and further elaborated into multiple applications in the medical field, such as orthopedics, dermatology, sports medicine and others, because of its capacity to retain growth factors and, thus, improve healing response at the application site.

Second-generation plasma concentrates consist of PRF and CGF. PRF was initially developed by Chakroun et al., using simple centrifugation for the application in surgical fields of dentistry, without using any additive materials, such as anticoagulants and thrombin [70,71]. The variation of these plasma-derived blood products is determined by two key predeterminants, which are the leukocyte volume and fibrin mass. The original derivatives, such as pure PRP, contain an immature, minute fibrin and fibrillae diameter, thus forming a less dense fibrin tissue adhesive. In general, these derivatives produce an unstable network and a high rate of tissue dissolution.

PRF consists of a stable, mature fibrin network, due to accomplished tissue polymerization accompanied by platelets and leukocytes forming a biomaterial with enhanced biomechanical tissue with structural integrity compared with the original PRP [71]. Nevertheless, CGF developed by Sacco produces an autologous membrane that is thicker, denser and more durable than the conventional PRF [72]. Ultimately, the demand and quest for soft- and hard-tissue healing response with optimal bone and tissue regeneration and remodeling are paramount. The plasma concentrates are still controversial with regard to their efficiency and effectiveness in integrating the bone–graft–implant–tissue complex, especially in the long term. However, the theoretical concepts of local application of growth factors will eventually enhance and support local healing and regeneration.

The structural integrity of APC ensures no tissue or wound breakdown, thus maintaining and promoting adequate angiogenesis and vascularity throughout the regenerative therapy and during clinical application. Isobe et al. compared the mechanical and biodegradable integrity of various types of APC, including CGF, A-PRF and platelet poor plasma-derived fibrin (PPTF), and showed that CGF and A-PRF exhibit almost similar properties under tensile strength and slower degradation than PPTF, thus theoretically limiting the potential usage of platelet poor plasma and its derivative in guided bone and tissue augmentation [73]. Panda et al. reported that the adjunctive use of APC, particularly PRF, shows better outcome in several parameters, including probing pocket depth, plaque index and clinical attachment level in root coverage periodontal infrabony defects, compared with the conventional GTR alone [74]. Miron et al. conducted a systematic review of 27 randomized clinical trials and found that the additive usage of PRF in comparison to open flap debridement alone improves radiographic bone fills and clinical attachment level [75]. Another recent study has also shown that adding PRF to CM with bovine xenografts results in a desirable outcome in alveolar ridge preservation, particularly in buccolingual width and vertical ridge height [76]. Fundamentally, APCs, particularly PRF, CGF and its derivatives, generate an excellent outcome when used as additives or to complement fundamental periodontal and regenerative procedures, such as open flap debridement and alveolar ridge preservation. Nonetheless, studies on the utilization of APC alone as a barrier membrane in the current literature search are lacking and, therefore, are recommended for future desirable works. The clinical example of utilization of APC is depicted in a series of photographs in Figure 3.

## 7. High-Performance Polymer

Polyetheretherketone (PEEK) is a semicrystallized thermosoftening polymer derived from the polyaryletherketone group. It is widely used in the medical field as an excellent alternative to titanium in orthopedics [77]. The research and application of PEEK in dentistry are extensive; specifically, it has been used as a dental implant, provisional abutment, obturator, denture base, clasp for dentures and others because of its good biological, mechanical, aesthetic and handling properties [78,79,80,81,82,83]. Given its excellent mechanical properties and structural integrity, PEEK has been suggested by Papia et al. to be used as a barrier membrane in complex three-dimensional surgery, because of its satisfactory mechanical properties under tensile and flexural strength with the thickness range of 0.5–1.0 mm, making it a desirable material for regeneration therapy [84]. In addition, it possesses the required general stiffness, strength and hardness, while maintaining ductility and light compared with other materials, such as polymer and ceramic [85,86]. The versatility of this material in manufacturing method, either milling under substractive computer-aided design and manufacturing (CAD-CAM) or rapid prototyping by additive manufacturing, makes it a favorable material to be utilized as a barrier membrane [87]. A study showed that the 3D bone augmentation utilizing a customized virtually designed PEEK sheet has satisfactory vertical and horizontal bone gain, with mean values of 3.47 and 3.42 mm, respectively [88]. A continuation study comparing the customized PEEK sheet and pre-bent titanium mesh achieved satisfactory outcomes in bone gain for both groups under CBCT assessment [89]. Additional clinical studies must be conducted to investigate the applicability of high-performance polymer PEEK or polyetherimide as a barrier membrane [90].

## 8. Conclusions

The advancement in science, particularly synthetic material and polymer, should be implemented in barrier membrane innovation for future improvement of the material. The addition of antimicrobial agents and properties complemented with nanomaterials technology should be incorporated to prevent potential complications during regenerative therapy and ensure the best possible outcome. Future improvements must focus on the advanced manufacturing in dentistry, including additive manufacturing and CAD-CAM for specific tailor-made computer engineered membranes for specific individuals and/or site needs. The quest for an ideal barrier membrane is dependent on the operator’s preference, skills and experience, rather than specific guidelines implemented for bone and tissue generation. The dentist should have in-depth knowledge of the material and techniques relevant to a specific barrier membrane to maximize the success of the procedure.

## Figures and Tables

**Figure 1 membranes-12-00444-f001:**
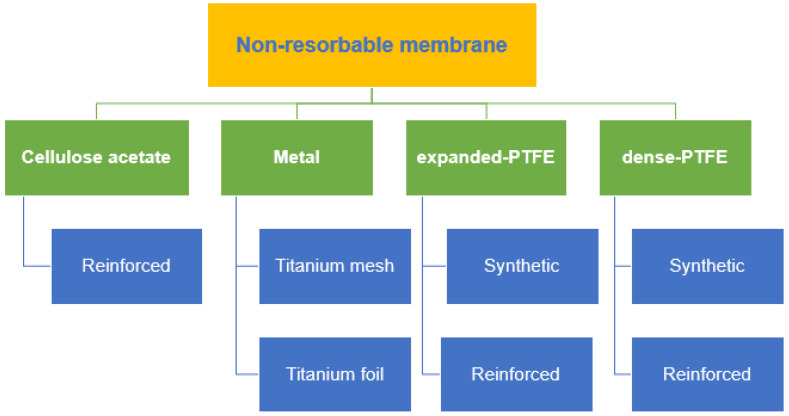
Various types of non-resorbable membranes.

**Figure 2 membranes-12-00444-f002:**
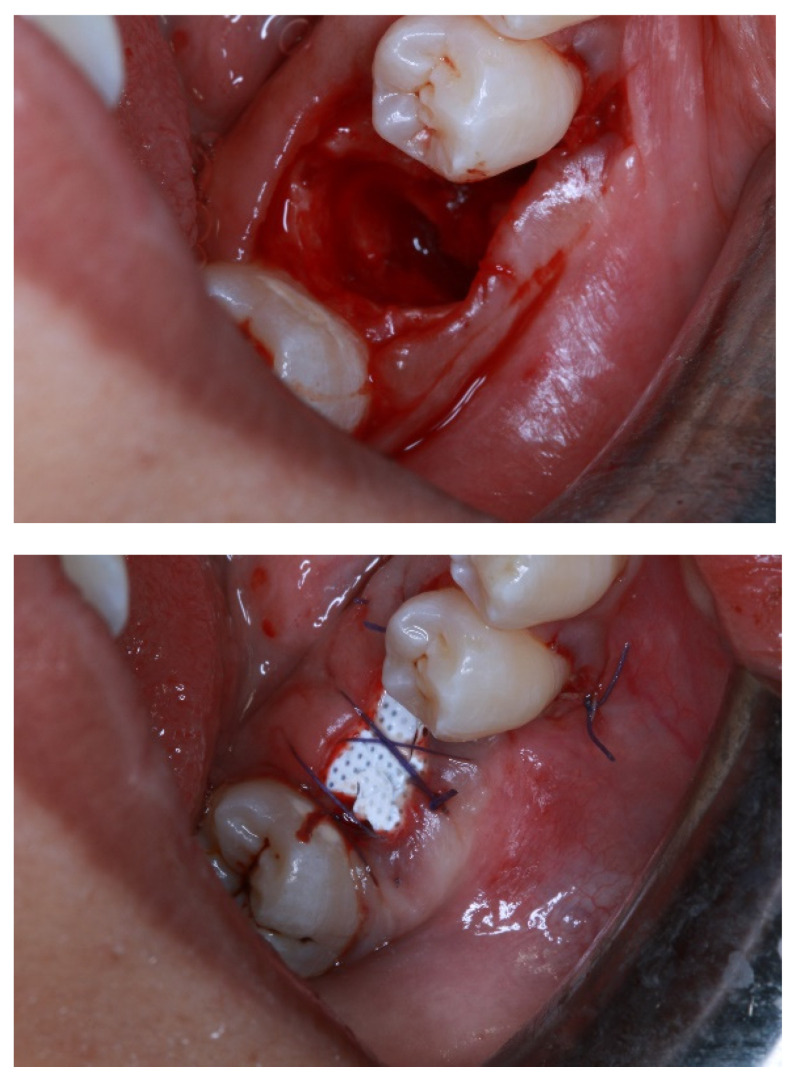
A schematic clinical photography series showing alveolar socket preservation performed on lower molar and utilization of dPTFE membrane secured by simple interrupted suturing technique. Note that the exposure of the barrier membrane is permissible, due to small microspores in properties, preventing bacterial ingress and, thus, simplifying surgical procedure. The final photo shows review of the surgical site upon removal of dPTFE membrane a month later.

**Figure 3 membranes-12-00444-f003:**
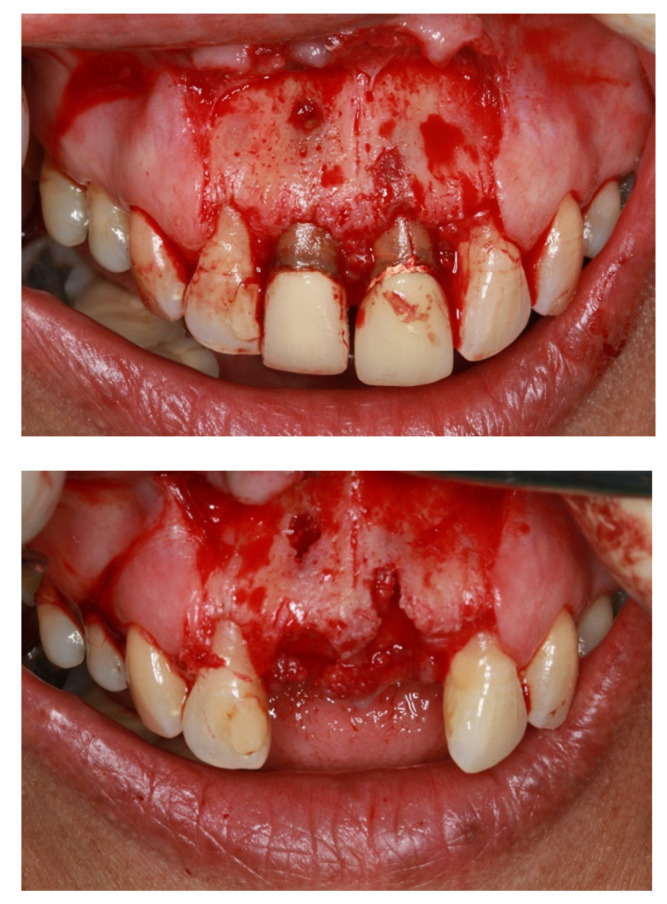
Pictographic series showing a surgical extraction performed to anterior teeth with alveolar ridge preservation, utilizing alveolar bone graft with simultaneous application CGF, which is part of APC. Note that the soft-tissue healed well after 3 weeks, and immediate provisionalization was performed.

**Table 1 membranes-12-00444-t001:** Commercially available non-resorbable barrier membrane.

Product (Company)	Material
Ti- Micromesh (ACE)	Titanium mesh
Tocksystem (MeshTM)	Titanium mesh
Millipore	Cellulose acetate
Gore-Tex^®^	ePTFE
Cytoplast™	dPTFE
Ti-Reinforced Gore-Tex^®^	Titanium-reinforced ePTFE
Cytoplast™ Ti-Reinforced 250	Titanium-reinforced dPTFE

## Data Availability

Not applicable.

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
