# Peer review of "Barrier Membrane in Regenerative Therapy: A Narrative Review"

_membranes, 2022, doi:10.3390/membranes12050444_

Round 1

Reviewer 1 Report

This paper reports a review about barrier membranes for Biomedical applications. It is recommended for publication in Membranes after major revision indicated below.

GENERAL COMMENTS

  • Rewrite the manuscript following the file format of the Journal.

SPECIFIC COMMENTS

  • It is important to remark the main novelty of this work.
  • Improve the quality of the figures.
  • Further possible studies related to the present work should be mentioned.
  • Future perspectives about the subject of study should be stated.
  • Please include more updated references.
  • Please include more references from the journal.

Author Response

A point-by-point response to the reviewer's comments had been elaborated in the attachment below

Reviewer 2 Report

The authors present a review on materials for regenerative therapy in dentistry, focusing on barrier membranes. Although the manuscript is overall well written and pleasant to read, it lacks of logical organization that makes it, sometime, hard to follow.

1. Major comments:

The format is misleading and should be adjusted. For example, from the format of the manuscript, I understood, that “i. Resorbable Membranes” was part of the “Methodology” section. Or in the Chitosan part, I do not understand the logic between the 1st and the 2nd §. Shouldn’t the 2nd § be in the collagen part? The 3rd and 4th § also seem to be misplaced…

Overall, the format should be adjusted with specific parts and sub-parts and the numbering should be consistent throughout the manuscript.

Also, I do not understand the logic of the growth factor part. It starts on the types of APC (it is actually the only sub-part) but APC does not seem to be a growth factor but a material (?)

2. Minor comments

-I regret the absence of figures to illustrate the examples cited in the text. Although not necessary, it would increase the manuscript attractiveness to the readers, and therefore, the probability to be cited.

-The manuscript should be edited for typos and missing words. Here are some examples:

Introduction part

2nd §:“to ensure progressive and complete integration with the periodontal fibres” should be fibers

Non-resorbable materials part

Page 11: “the adaptation and fixation of the device to the recipient site time consuming [48]” – the verb is missing.

Type of APC part

3rd §: “Sacco produces an autologous membrane that is thicker, denser and durable than the conventional PRF” isn’t it “more durable”?

3rd §: “The plasma concentrates are still controversial with regard their efficiency […]”. It should be either “regarding their […]” or “with regard to their […]”.

- I noted one missing acronym’s definition

Growth factors part

APC is not defined.

Author Response

A point-by-point response to the reviewers's comments had been elaborated in the attachment file below

Reviewer 3 Report

This review article is insightful to the readers. Also, it is well organized and scholarly written. So, it can be accepted for publication. 

Author Response

a point-by-point response to the reviewers's comments had been elaborated in the attachment below 

Round 2

Reviewer 1 Report

Authors carried out all the changes suggested during the previous review round.

Author Response

Please see the attachment for the amendment.

Thank you

Reviewer 2 Report

I would like to thank the authors for considering my comments.

In the latest version, I would just underline a small error in the part number (#7 is missing) and on page 9, 'Synthetic Membrane' is bold. It should either be underlined or have a number, depending on whether it is a new part or a sub-part.

You might also want to combine Fig2 and 3, to be consistent with Fig4.

All the best for this paper

Author Response

Please see the attachment for the amendment.

Thank you.
